# Spatiotemporal coupling and coordinated development of rural revitalization and rural tourism in Jiangsu

Meiqin Ding *¤, Hui Liu

School of Air Transport and Engineering, Nanjing University of Aeronautics and Astronautics Jincheng College, Nanjing, Jiangsu, China

¤ Current address: School of Air Transport and Engineering, Nanjing University of Aeronautics and Astronautics Jincheng College, Nanjing, Jiangsu, China
* alading0827@nhjcxy.edu.cn

## Abstract

Rural tourism is pivotal in addressing the unidirectional urban-to-rural flow of resources, such as labor migration. However, the interaction between rural tourism and rural revitalization in developed regions remains poorly understood. This study establishes an evaluation index system for rural revitalization and rural tourism, examining their interrelationship. Using the entropy method and coupling coordination degree model, we assess the development levels and coordination degrees of these aspects in Jiangsu Province from 2012 to 2023. Furthermore, the geographical detector model is utilized to pinpoint the primary drivers influencing this coordination. The findings are: (1) Both rural revitalization and tourism exhibit significant growth, with southern Jiangsu outperforming the north; (2) The coupling coordination between these systems has strengthened, indicating a profound symbiotic relationship; (3) Spatial distribution differences are notable, with the coupling coordination degree D value in southern Jiangsu being 26.4% higher than in the north. This disparity is primarily attributable to the wider urban-rural income gap and greater fiscal investment in southern Jiangsu. Notably, the traditional "resource dependence theory" appears ineffective in Jiangsu, as the density of rural tourism resources is relatively low. Accordingly, the study proposes differentiated policy recommendations: northern Jiangsu should focus on talent attraction and the integration of culture and tourism, while southern Jiangsu should explore mechanisms to facilitate the two-way flow of urban-rural elements. This research provides a theoretical framework for coordinating "policy-market" dynamics in the rural transformation of developed regions.

## 1. Introduction

Amid the growing focus on global urban-rural integration and rural revitalization strategies, rural tourism has emerged as a pivotal component in advancing rural

**Data availability statement:** Some of the data used in this article can be obtained from the following websites and publications, including: 1.Jiangsu Statistical Yearbook (2013-2024) (https://tj.jiangsu.gov.cn/col/col91733/index.html) 2.Jiangsu Yearbook (2013-2024) 3.Statistical Bulletin on National Economic Development of Jiangsu Province (2012-2023) (https://tj.jiangsu.gov.cn/col/col87586/index.html) 4. Statistical Database of China Economic Network (https://ceidata.cei.cn) 5. Official websites of central and local governments (such as Ministry of Agriculture and Rural Affairs, Forestry Bureau of Jiangsu Province, Department of Ecological Environment of Jiangsu Province, official websites of prefecture-level cities, etc.) 6. Others (such as China Civilization Network, China Ethnic Culture Resource Library, China Environmental Statistics Yearbook, etc.) There is also a part of the data, because it is obtained from government platform letters or telephone consultation, belongs to non-public data, can not be publicly shared data. Data can be obtained from consultation with government departments of Jiangsu Province or prefecture-level cities. Because there are many relevant data sources, the linked list of the paper dataset is uploaded as a supporting information file as supplementary information to the data availability statement. Please consult the relevant documents.

**Funding:** This research was funded by the General Project of Philosophy and Social Science Research in Jiangsu Provincial Universities (2022SJYB0714). The funders had no role in study design or data.

**Competing interests:** The authors have declared that no competing interests exist.

revitalization [1], has attracted great attention from policymakers. Currently, the academic community has widely recognized the interrelationship between rural revitalization and rural tourism, examining the latter's development within the context of the former. However, existing studies mostly rely on interface data, making it difficult to capture dynamic evolution. Notably, the quantitative analysis of high-density ruralization's impact on the coupling mechanism in developed regions remains unexplored. Thus, a thorough investigation into achieving the seamless integration of these elements is urgently required.

With the acceleration of the global urbanization process, a large – scale migration of the rural population to cities has occurred. Villages see labour shortages, recession and social degradation. In most regions, countryside communities are becoming less viable [2]. According to the data from the World Bank's 2023 report, the rural population loss rate in developing countries reached 37% from 2000 to 2020. In response, various countries have implemented initiatives aimed countermeasuresat rural revitalization, such as Germany's "Village Renewal" project, Japan's "One Village, One Product Movement and Rural Revitalization Initiative", and South Korea's "New Village Movement" [3,4]. Different from the private land ownership in Europe, China implements the collective land ownership system and promotes rural construction in an orderly manner through a government – led model. This also brings certain challenges to rural development, namely the one – way flow of urban – rural elements. Rural labor and funds flow to cities, while there is less back – flow of urban resources, resulting in unbalanced rural development. Thus, China started rural development version 2.0, namely rural revitalization (RR). It refers to the national strategy that focuses on agriculture, rural areas, and villagers and promotes the comprehensive and coordinated growth of the rural economy, ecology, culture, and society to narrow the urban – rural gaps gradually and achieve shared prosperity [5].

Rural tourism is a new type of ecological industry that integrates the primary, secondary, and tertiary industries, with agriculture as the foundation, tourism as the purpose, and services as the means. Due to its strong linkage and driving effect, it can effectively guide and promote the flow of more resources and factors to rural areas, bringing profound impacts on the development of rural areas in China. Studies have shown that rural tourism promotes rural economic growth [6], enriches employment opportunities [7], and improves entertainment and cultural facilities [8]. Meanwhile, rural revitalization can not only create a superior natural environment foundation and a solid economic backing for the booming development of rural tourism but also play a key role in the high – quality development of rural tourism. Therefore, there is a natural coupling relationship between rural revitalization and rural tourism. China has a vast territory, and the issue of unbalanced rural development has long existed [1]. Previous research has primarily centered on impoverished areas [9,10], neglecting the reshaping of the coupling relationship by the 'high - density rurality' in developed areas (e.g., the number of scenic spots per 100 square kilometers in southern Jiangsu is 7.2, while the national average is 2.1). Through constructing a dynamic panel model (2012–2023) and a multi – scale geographical detector, this study reveals for the first time: (1) the non-linear transition law of the coupling coordination

degree; (2) the 'institution - resource' complementary mechanism between southern and northern Jiangsu. These findings provide a Chinese paradigm for rural transformation in global developed economies (such as the Ruhr area in Germany and villages in Japan).

## 2. Literature review

### 2.1 Definition and essence of rural tourism

Rural tourism (RT) can be regarded as a phenomenon that can be traced back to the end of the 19th century [11,12]. RT can be viewed as a phenomenon resulting partly from the wish to escape the urban environment and the need to reaffirm personal identities in the face of growing urbanisation [13]. Rural areas have a special appeal to tourists because of the mystique associated with rural areas and their distinct cultural, historic, ethnic, and geographic characteristics. Meanwhile, with the process of globalization and urbanization, rural areas have been undergoing transformation in social and economic aspect [1]. Rural tourism is often described as a means to restore socio – economic development [14]. Although there is no consensus in the academic community on the definition of rural tourism, the most frequently cited one is that of Lane (1994). Lane (1994, p. 14) defined RT as "The tourism which satisfies these forms: located in rural areas, functionally rural, set in rural scale, traditional in character, representing the complex pattern of rural environment, economy, history and location [15]." As most authors define RT by describing key tourism activities in rural destinations [16], such as farm – based tourism, nature – based tourism, cultural tourism, agrotourism, ecotourism and other related activities in rural area [17,18]. Additionally, the impact of digital technologies, such as online platforms and short videos, is being investigated for their potential to transform RT's operational modes and experiences [19,20].

Rurality is generally deemed as the essence of rural tourism development [21]. In rural tourism destinations, rurality is the core that differentiates them from urban tourism settings and is regarded as rural idyllic or rural nostalgia [22]. As the essential characteristic of rural tourism, rurality is a key indicator for measuring and judging the development of rural tourism [23]. Traditional studies often define rurality through population, location, and landscape [1], with limited exploration of its specific dimensions and variations across different development levels and cultural contexts, such as in high-density, developed regions. In areas like southern Jiangsu, China, rural tourism is marked by policy intervention, high population density, and urban-rural integration, raising questions about whether current definitions capture its true essence. Rapid urbanization and rural revitalization are transforming "rurality", challenging the assumption that rural characteristics are the core of rural tourism [24]. As tourist expectations rise, existing research lacks depth on whether landscapes shaped by tourists and developers retain authentic rurality. In economically advanced regions like Jiangsu, where urban-rural distinctions blur, redefining rurality to reflect "high-density rural development" is increasingly necessary.

### 2.2 Rural tourism as an effective approach to rural revitalization

With the rapid advancement of urbanization and industrialization, rural decline has gradually become a major challenge faced by countries all over the world [25,26]. Rural areas around the world are confronted with challenges such as a decline in economic activities, adjustment of the traditional agricultural industrial structure, population aging, the outflow of highly – educated youth, and a decrease in the viability of small towns and villages [15,27,28]. To address this challenge, countries around the world have actively implemented a series of policies and measures aimed at promoting rural reconstruction and revitalization in combination with their specific local conditions. Different from the European Union, which promotes community – led development through the "LEADER Program" [29], rural revitalization in China relies more on government – led "area - based development" (e.g., Suzhou).

With the decline of agriculture, rural development strategies have begun to emphasize the diversification of economic activities [30]. Among them, Tourism presents significant potential in fostering social transformation in rural areas (UNWTO, 2023). Rural tourism is widely regarded as an effective tool for stimulating rural economic growth [6,31,32] and

revitalizing local cultural identities [33,34]. A series of policies and initiatives by the Chinese government show unprecedented support for developing rural tourism [35]. While the potential benefits of rural tourism (RT) are acknowledged, there remains a lack of in-depth understanding regarding the specific pathways through which it achieves multidimensional rural revitalization (e.g., industrial, ecological, cultural) via mechanisms such as factor flow and industrial linkage.

Excessive or poorly managed tourism development can have detrimental effects. For example, Dai (2023) observed that "Although tourism development seems to have achieved the economic and governmental goals of the revitalization strategy, the living environment and social etiquette and civility have been destroyed" [36]. Therefore, to maintain the sustainability of rural revitalization, different needs must be addressed [11], such as rural environmental safety [37], taking into account rural daily life while promoting rural development [38], and the inheritance of rural culture [36]. Prior research indicates that the primary reason or rationale for government intervention in tourism has transformed from mainly economic considerations to the extended concerns over environmental and social consequences of tourism development [39,40]. Additionally, scholars have observed that the intensity of policy intervention may affect market vitality [41]. However, the optimal conditions for policy intervention to effectively enhance the positive synergy between rural tourism and rural revitalization, including influence thresholds and context dependencies like resource endowment and development stage, require further study.

### 2.3 Application and limitations of the coupling and coordination theory

Coupling refers to the phenomenon of interaction, mutual influence and constant synergy between two or more systems [42]. The coupling degree describes the extent of interaction and influence between different systems, and the coupling coordination degree represents the magnitude of the benign coupling degree of coordination between systems [43]. Society is a complex system, and there are multiple coupling relationships between different systems such as resources, ecology, economy and society [44,45]. The coupling coordination relationship is a positive association between different systems. When the coupling coordination degree between them is high, it indicates a benign development relationship between the systems [46,47]. Meanwhile, they dynamically change over time [48]. The coupling coordination model can avoid the influence of subjective factors.It has a certain objectivity and universal applicability [49], and its effectiveness has been confirmed in successful applications in industries, such as energy, tourism, social field [50,51].

Recent studies have confirmed the intrinsic link between rural revitalization and rural tourism [52], with some researchers employing the coupling coordination model for analysis [10].However, these studies often treat the research area as a homogeneous entity, focusing on evaluating and comparing the degree of coupling coordination and its spatio-temporal distribution characteristics. They neglect the spatial differentiation of the coupling coordination degree and the fundamental differences in its driving forces, which can be attributed to factors such as economic foundation, resource endowment, policy intensity, and urban-rural relations within the region. Consequently, recommendations for coordinated development based on these findings may lack persuasiveness. Furthermore, while some scholars have verified the coupling and linear relationships between rural revitalization and rural tourism using linear regression model [9], they have failed to uncover the non-linear characteristics (e.g., critical points, threshold effects, and path dependence) and feedback mechanisms between the two systems. This limitation restricts the application scope and explanatory power of the existing models to a certain extent [53].

### 2.4 Research gaps and the positioning of this paper

While the relationship between rural revitalization and tourism has been extensively studied, empirical analysis of the long-term, nonlinear dynamics of their coupling and coordination remains limited. Research indicates significant spatial differences in this relationship within Jiangsu, with stronger coupling in the south compared to the north. The specific driving factors and spatial interaction mechanisms contributing to these differences are not well understood, and traditional homogenized models fall short of explaining them. In densely developed regions like Jiangsu, where resource

endowments are less prominent, the role of high-level policy interventions and urban-rural synergy in sustaining high coupling and coordination levels is unclear. This raises questions about the applicability of the "resource dependence theory" [54] and highlights a critical gap in both theoretical understanding and practical application. Furthermore, variations in research perspectives have hindered the full development of evaluation indices for existing coupling models. Objectively determining the weights of these indices to accurately reflect their coupling and coordination remains a critical methodological challenge.

Addressing existing research gaps, this paper introduces innovations by analyzing panel data from Jiangsu Province (2012–2023). We develop a multi-dimensional evaluation system, weighted via the entropy method, and employ the coupling coordination degree model to elucidate the dynamic evolution and spatiotemporal characteristics of RR and RT. Utilizing the geographical detector, we investigate the driving mechanisms of spatial heterogeneity. The analysis focuses on testing whether and how "policy intervention" and "urban-rural synergy" can overcome "resource constraints" to foster a high-level coupling coordination pathway, amidst the backdrop of "high-density ruralization" in developed regions. In doing so, the study challenges resource-dependence theory and offers novel theoretical perspectives and differentiated policy foundations for rural transformation in developed regions.

The proposed conceptual model (Fig 1) suggests a reciprocal relationship between rural revitalization and rural tourism development. Based on the above analysis, the information graphics that concern the research framework are shown in Fig 2.

## 3. General situation of the study area, research methods, and data sources

### 3.1 Overview of the study area

Jiangsu Province, with its robust economy, has a thriving rural tourism sector. It features 7 national key rural tourism towns, 53 national key rural tourism villages, and 142 provincial key rural tourism villages, each offering distinct attractions. Southern Jiangsu prioritizes high-quality rural tourism services with a focus on local culture, central Jiangsu concentrates on improving product quality, and northern Jiangsu emphasizes the development of rural sightseeing products. The

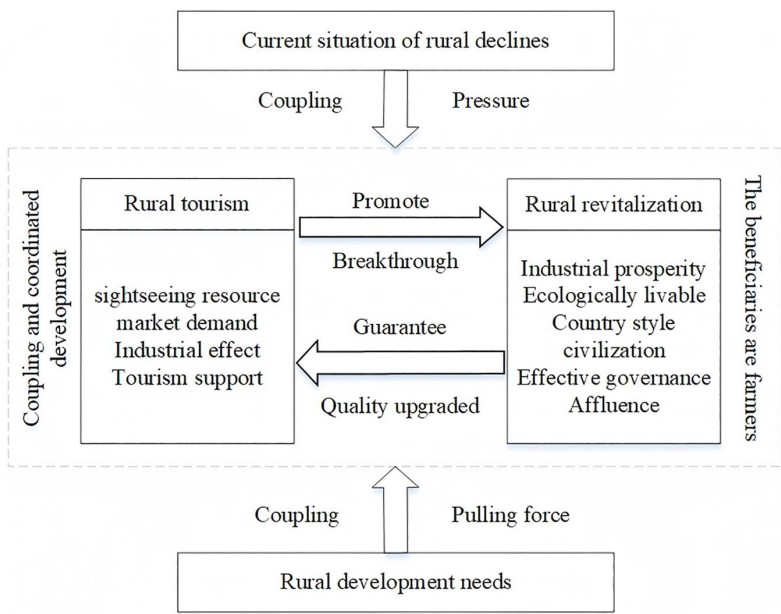

**Fig 1. Coupling relationship between rural revitalization and rural tourism.**

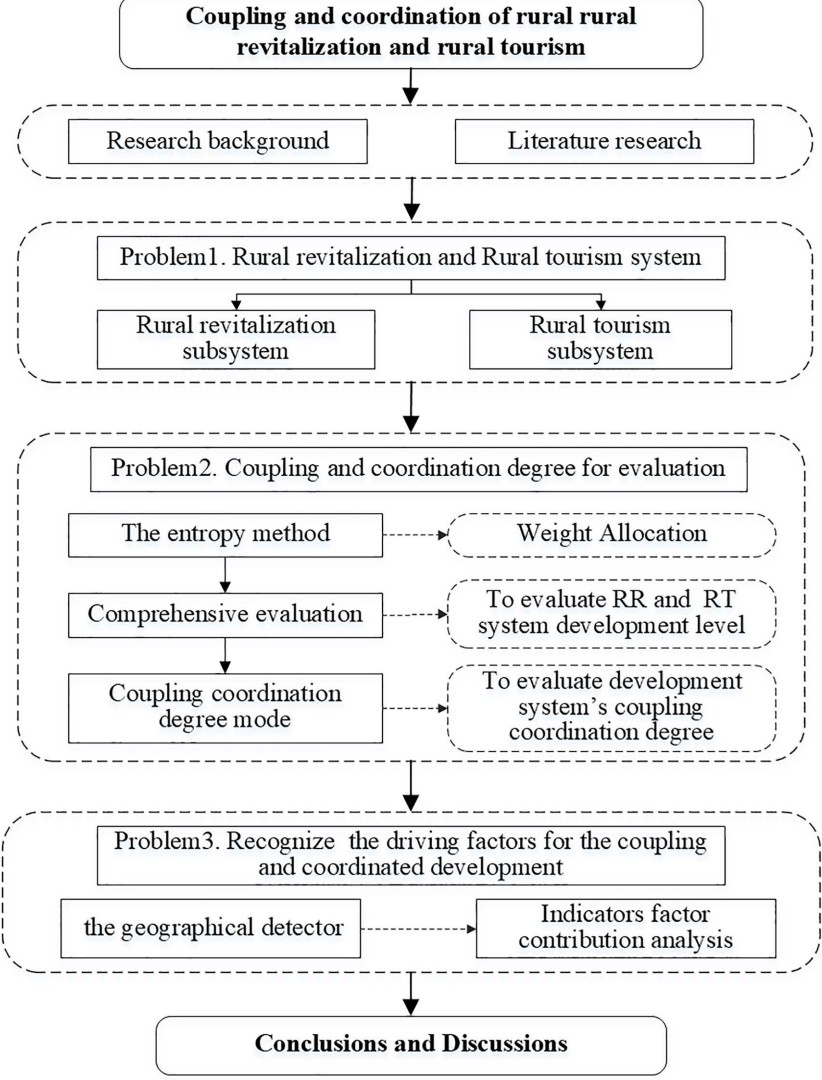

**Fig 2. Graphical information of the research framework.**

provincial government's "Three-year Action Plan for Rural Tourism Development" (2015) and "Guidelines for Rural Tourism Development" (2018) have outlined strategies, objectives, policies, and support mechanisms, significantly boosting the industry's quality and efficiency.

## 3.2 Construction of the evaluation index system for rural revitalization – rural tourism

Drawing on the "Rural Revitalization Strategy Plan (2018 - 2022)" [55], the "Guidelines for the Construction of Beautiful and Livable Villages" [56],and insights from other regions, this paper thoroughly examines the core elements of the evaluation framework for rural revitalization and tourism(Table 1). The rural revitalization model includes five main indicators and sixteen sub-indicators, concentrating on industry, ecology, customs, governance, and quality of life. Meanwhile, the rural tourism model features four primary indicators and nine sub-indicators, focusing on resources, market demand, industry impact, and support services.

**Table 1. Coupling evaluation index system for rural revitalization and rural tourism in Jiangsu Province.**

| System | First-level indicators | Secondary indicators | Unit | Indicator direction | References |
|---|---|---|---|---|---|
| Rural revitalization | Flourishing industries | Agricultural labor productivity | Ten thousand yuan per person | Positive | [57] |
| | | Gross output value of agriculture, forestry, animal husbandry and fishery | 100 million yuan | Positive | [58] |
| | | Total power of agricultural mechanization | Ten thousand kilowatts | Positive | [59] |
| | Ecologically livable | Forest coverage rate | % | Positive | [60] |
| | | Ratio of days with good air quality | % | Positive | [60] |
| | | Popularization rate of harmless sanitary toilets | % | Positive | [60,61] |
| | Rural cultural civilization | The number of national civilized villages and towns | Unit | Positive | [58] |
| | | The number of intangible cultural heritages at or above the provincial level | Unit | Positive | [62] |
| | | The number of township cultural stations | Unit | Positive | [58] |
| | | The proportion of household expenditure on culture, education, and entertainment | % | Positive | [59] |
| | Effective governance | Number of people receiving minimum living security | Ten thousand people | Negative | [58,63] |
| | | Rural employees | Ten thousand people | Positive | [58] |
| | | Urban-rural residents' income ratio | % | Negative | [64] |
| | Affluent life | Per capita disposable income of farmers | Ten thousand yuan | Positive | [59,62] |
| | | Engel's coefficient of rural residents | % | Negative | [59,62] |
| | | Per capita floor area of current housing in rural areas | Square meters | Positive | [59,62] |
| Rural tourism | Tourism resources | National Leisure Beautiful Villages | Unit | Positive | [65] |
| | | Traditional ancient villages in China | Unit | Positive | [66] |
| | | Leisure agricultural parks and scenic spots of a certain scale | Unit | Positive | [67] |
| | Market demand | Total number of tourists | 100 million person – times | Positive | [62,65,68] |
| | | Tourism revenue/ Comprehensive revenue of leisure agriculture | 100 million yuan | Positive | [62,65,68] |
| | Industrial effect | Number of tourism practitioners/ Number of practitioners in leisure agriculture | Ten thousand people | Positive | [62] |
| | | The proportion of tourism revenue in GDP | % | Positive | [62] |
| | Tourism support | Highway passenger turnover volume | 100 million passenger – kilometers | Positive | [58] |
| | | The number of travel agencies | Unit | Positive | [58,67] |

Note: The data sources of each indicator in this table can be found in the "References" column, and the corresponding complete literature information is listed in the references [57–68] at the end of the text.

## 3.3 Data sources

This study analyzed 25 evaluation indicators in Jiangsu Province from 2012 to 2023. The data utilized in this study were primarily sourced from the following: Agricultural and farmer-related data, as well as highway passenger turnover volume and travel agency statistics, were obtained from the Statistical Yearbook of Jiangsu Province and the statistical yearbooks of individual cities. Forest coverage rates were gathered from the official website of the Jiangsu Forestry Bureau and the Statistical Bulletin on the National Economic Development of Jiangsu Province. The rate of days with good air quality in each prefecture-level city was collected from local government websites. The popularization rate of harmless sanitary toilets was sourced from the China Environmental Statistical Yearbook and the websites of prefecture-level city

governments. The number of national civilized villages and towns was obtained from the official website of the Central Spiritual Civilization Office. Intangible cultural heritage data were retrieved from the China National Cultural Resources Database. National leisure beautiful village data were gathered from the official website of the Ministry of Agriculture and Rural Affairs. Additional data were primarily acquired through information consultation on the Jiangsu Yearbook and the government information consultations of various cities. For missing individual indicator data, interpolation methods were employed to supplement and improve the dataset.

## 4. Research methods

### 4.1 Determine the index weights

**4.1.1 Data standardization processing.** To eliminate the disparate orders of magnitude in the original data, we employed the range method to standardize the values. Additionally, to preclude the inclusion of invalid data, we uniformly shifted the standardized values by 0.0001 to the right.

$$\text{Positive indicators}: X'_{ij} = \frac{X_{ij} - \min X_j}{\max X_j - \min X_j} + 0.0001 \tag{1}$$

$$\text{Negative indicators}: X'_{ij} = \frac{\max X_{ij} - X_{ij}}{\max X_j - \min X_j} + 0.0001 \tag{2}$$

**4.1.2 Calculate the weights.** This study employs the entropy method to assign weights to each indicator. This method can effectively avoid the subjectivity and arbitrariness brought by subjective weighting methods and have been widely applied in multiple disciplines [69,70]. The precise calculation is outlined in Equations (3)-(6).
① Calculate the proportion of the j-th indicator in the i-th year:

$$p_{ij} = \frac{X'_{ij}}{\sum\limits_{i=1}^{n} X'_{ij}} \tag{3}$$

② Calculate the entropy value of the j-th indicator:

$$e_j = -k \sum\limits_{i=1}^{n} p_{ij} \ln p_{ij} \tag{4}$$

③ Calculate the coefficient of variation for the j – th indicator:

$$U_j = 1 - e_j \tag{5}$$

④ Determine the weight of the j-th indicator:

$$W_j = \frac{u_j}{\sum\limits_{i=1}^{m} u_j} (1 \leq j \leq m) \tag{6}$$

## 4.2 Comprehensive evaluation of multiple indicators

After determining the weights of each indicator, calculate the comprehensive evaluation value of the i-th evaluation indicator. The calculation formula is as follows:

$$S = \sum_{j=1}^{m} (w_j * X'_{ij})$$

(7)

## 4.3 Coupling coordination degree mode

This study employs a coupling coordination model to assess the interaction intensity and coordination level between rural revitalization and rural tourism systems. The calculation steps are as follows:

$$C = 2 \left[ \frac{U \times G}{(U + G)^2} \right]^{\frac{1}{2}}$$

(8)

$$T = \alpha \times U + \beta \times G$$

(9)

$$D = (C \times T)^{\frac{1}{2}}$$

(10)

In Eqs. (8)~ (10), C denotes the coupling degree, T is the comprehensive coordination function of the two systems, and D signifies the coupling coordination degree. U and G are the comprehensive evaluation values for the levels of rural revitalization and rural tourism development, respectively. The weights of the two major systems are represented by α and β, and in this study, they are considered to be of equal importance, with α = β = 0.5. Based on previous research [71,69], combined with the actual situation of this study, the coupling degree and coupling coordination degree can be divided into different grading standards, as shown in Table 2.

## 4.4 Geographical detector

Geographical detector, a statistical method for identifying spatial heterogeneity and its driving forces [72], effectively elucidates the formation mechanisms of spatial distributions without prior assumptions. Its application spans various socio-economic fields. This study employs the geographical detector to examine the spatial differentiation in the coupling and coordinated development of rural revitalization and tourism, identifying the influencing factors. The model is presented as follows:

$$q = 1 - \frac{\sum_{h=1}^{L} N\sigma_h^2}{N\sigma^2} = 1 - \frac{SSW}{SST}$$

(11)

In Equation (11), q quantifies the explanatory power of the driving factor X, ranging from to 1. A higher q value signifies a stronger influence of X on the coordination degree.

## 5. Spatiotemporal coupling of rural revitalization and rural tourism in Jiangsu Province

The evaluation index system was standardized using established formulae. Entropy analysis and relevant models were employed to calculate the comprehensive evaluation scores and coupling coordination degrees, which characterized

**Table 2. Coupling degree and coordination degree classification.**

| C values | Coupling level | D values | Coordination level |
|---|---|---|---|
| [0, 0.3] | Low-level coupling | [0, 0.1] | Extreme maladjustment |
| | | (0.1, 0.2] | Serious maladjustment |
| (0.3, 0.5] | Antagonistic coupling | (0.2, 0.3] | Moderate maladjustment |
| | | (0.3, 0.4] | Mild maladjustment |
| | | (0.4, 0.5] | On the verge of maladjustmen |
| (0.5, 0.8] | Adaptive coupling | (0.5, 0.6] | Basic coordination |
| | | (0.6, 0.7] | Primary coordination |
| | | (0.7, 0.8] | Intermediate coordination |
| | | (0.8, 0.9] | Good coordination |
| | | (0.9, 1] | Excellent coordination |

the development levels of rural revitalization and rural tourism in Jiangsu Province from 2012 to 2023, as well as in its 13 prefecture-level cities in 2023. The coupling coordination relationship between these two systems was subsequently analyzed.

## 5.1 Comprehensive evaluation of the development levels of rural revitalization and rural tourism

### 5.1.1 Analysis of the temporal characteristics of the comprehensive development level in Jiangsu Province.
Fig 3 shows the synchronous growth of rural revitalization and tourism in Jiangsu Province, with average annual increases of 22.64% and 31.7%, respectively. This trend is largely due to the pivotal role of local governments in these sectors in rural development and rural tourism [73]. Over the past decade, Jiangsu has focused on developing distinctive rural landscapes and advancing revitalization through targeted regulations. Key initiatives include the development of modern agriculture since 2008, the "Beautiful Urban-Rural Construction Campaign" in 2011, the proposal of characteristic idyllic villages in 2017, and the rural revitalization strategy in 2018. Meanwhile, the Provincial Government has implemented several support policies for rural tourism, including the Three-year Action Plan (2016–2018), Guidelines for Rural Tourism Development (2018–2023), and Guiding Opinions on High-quality Development (2021), have established clear objectives, driving significant momentum in rural tourism growth.

Between 2012 and 2018 in Jiangsu Province, the level of rural revitalization in Jiangsu Province remained high, aligning with the findings of Li, Long, and Li [23]. During this period, rural tourism expanded more rapidly. As Minister Han from

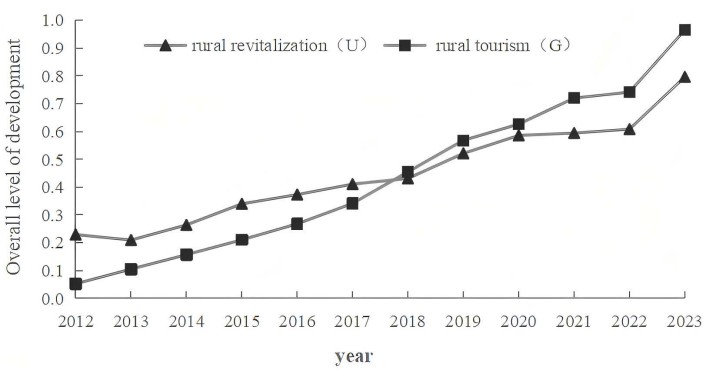

**Fig 3. Comprehensive evaluation scores of rural revitalization and rural tourism in Jiangsu Province from 2012 to 2023.**

the Ministry of Agriculture (2017) highlighted, the development of leisure agriculture and rural tourism should stand on the basis of agriculture, which indicates that rural revitalization exerts a substantial influence on rural tourism [74]. From 2019 to 2023, rural tourism overtook rural revitalization, emerging as a primary catalyst for the latter [75]. This transition facilitated a phase of coordinated development between the two sectors.

**5.1.2 Spatial pattern characteristics of the comprehensive development level in Jiangsu Province.** Cities in Jiangsu Province are classified into seven tiers based on development indicators, spanning from very weak to very strong. In 2023, most prefecture-level cities, excluding Nanjing, Suzhou, and Yancheng, demonstrated underdeveloped rural tourism. As shown in Fig 4, a clear imbalance exists in regional and urban development, particularly in rural tourism, with southern Jiangsu outperforming the central and northern areas.

The rural revitalization values across prefecture-level cities in Jiangsu Province exhibit significant regional disparities. As mentioned by Chen, Y., et al. (2025), the factors affecting the rural development level include resource endowment, economic development level, education level, and government support for agriculture [76]. Southern Jiangsu, with a stronger economic base, demonstrates greater investment and faster progress in rural revitalization. In northern Jiangsu, Xuzhou and Yancheng also exhibit high development levels, which can be attributed to the prosperity of their agricultural industries. Notably, rural tourism development is most advanced in Nanjing and Suzhou, with Yancheng closely following. These advancements are driven by a combination of market dynamics and government policies. Nanjing and Suzhou benefit from robust urban economies and tourism demands, supported by government policies and infrastructure. Yancheng's success is linked to its rich rural tourism resources, following a tourism-driven development model. In contrast, rural tourism in other Jiangsu cities lags behind.

The uneven development of rural revitalization and tourism across prefecture-level cities in Jiangsu Province could widen regional disparities in rural economic and social progress. Consequently, government agencies must persistently

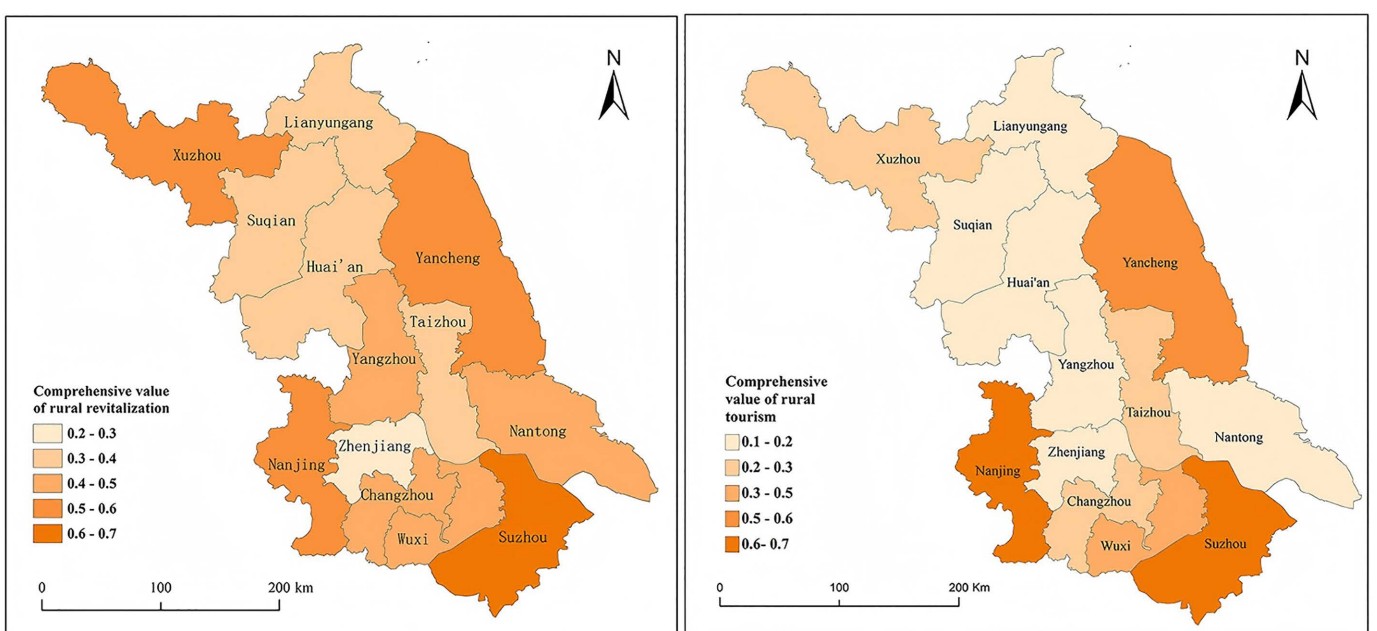

**Fig 4. Spatial pattern of the comprehensive development levels of rural revitalization and rural tourism in 13 cities of Jiangsu Province in 2023.** The map is based on the standard map with review number GS (2024) 0650 downloaded from the Standard Map Service website of the Ministry of Natural Resources (http://bzdt.ch.mnr.gov.cn/), with no modifications to the base map.

refine policy guidance and investigate new mechanisms for cross-regional collaboration to bolster the overall strength and market competitiveness of rural revitalization and tourism in the province.

## 5.2 Temporal coupling between rural revitalization and rural tourism

The coupling between rural revitalization and rural tourism in Jiangsu Province shows certain phased characteristics in terms of time (Table 3).

(1) From 2012 to 2023, the coupling degree curve of rural revitalization and tourism in Jiangsu Province reveals an "L-shaped" leap, signifying a deeply embedded symbiosis. The high coupling level (C ≥ 0.943) after 2012, except initially, stems from the province's early and well-developed rural tourism sector. Jiangsu's sustained growth and quality improvements in rural tourism have propelled regional revitalization goals. A decade of rural development initiatives and comprehensive policy support have laid a solid foundation for the synergistic advancement of rural revitalization and tourism.

(2) The coupling coordination between rural revitalization and rural tourism in Jiangsu Province exhibited a steady upward trajectory from 2012 to 2023. The coupling degree, as measured by the D-value, with an average annual growth rate of approximately 9.8%. This transition reflects a progression from an inefficient interaction stage in 2012 to a high-efficiency collaboration stage by 2023. The evolution of this coupling coordination followed a "three-stage ladder-type" growth pattern. The initial 2012–2014 period marked a stage of weak system coordination potential, characterized by poor coordination between subsystems, mild imbalance, and a lack of effective synergy due to inefficiencies in rural policies and the tourism market.From 2015 to 2020, a dynamic balance emerged between rural revitalization and tourism, evolving from loose to moderate coordination. This era saw increased interaction, tighter coupling, and enhanced synergy between the sectors, fostering rural industry integration. By 2021–2023, a high-order collaboration was attained, with improved coordination achieving a balanced development. In 2023, surpassing the critical value of 0.93 indicated a leap towards high-quality coordination. This established a positive cycle of "promoting tourism with agriculture and boosting agriculture with tourism" in Jiangsu Province's rural areas.

Rural tourism in Jiangsu Province has experienced substantial and sustained growth. The integration of rural tourism and rural revitalization initiatives has advanced, exhibiting enhanced synergy and coordination. Future efforts should prioritize deepening this integration, implementing supportive policies, upgrading the rural tourism industry, enhancing service

Table 3. Coupling coordination degree between rural revitalization and rural tourism in Jiangsu Province.

| Year | Coupling degree C | Coupling coordination degree D | Coordination type |
|------|-------------------|--------------------------------|-------------------|
| 2012 | 0.7782 | 0.3312 | Mild dysregulation |
| 2013 | 0.9435 | 0.3854 | Mild dysregulation |
| 2014 | 0.9672 | 0.4512 | On the verge of dysregulation |
| 2015 | 0.9720 | 0.5171 | Forced coordination |
| 2016 | 0.9864 | 0.5622 | Forced coordination |
| 2017 | 0.9957 | 0.6117 | Primary coordination |
| 2018 | 0.9996 | 0.6652 | Primary coordination |
| 2019 | 0.9991 | 0.7374 | Intermediate coordination |
| 2020 | 0.9994 | 0.7780 | Intermediate coordination |
| 2021 | 0.9954 | 0.8085 | Good coordination |
| 2022 | 0.9951 | 0.8191 | Good coordination |
| 2023 | 0.9954 | 0.9360 | High-quality coordination |

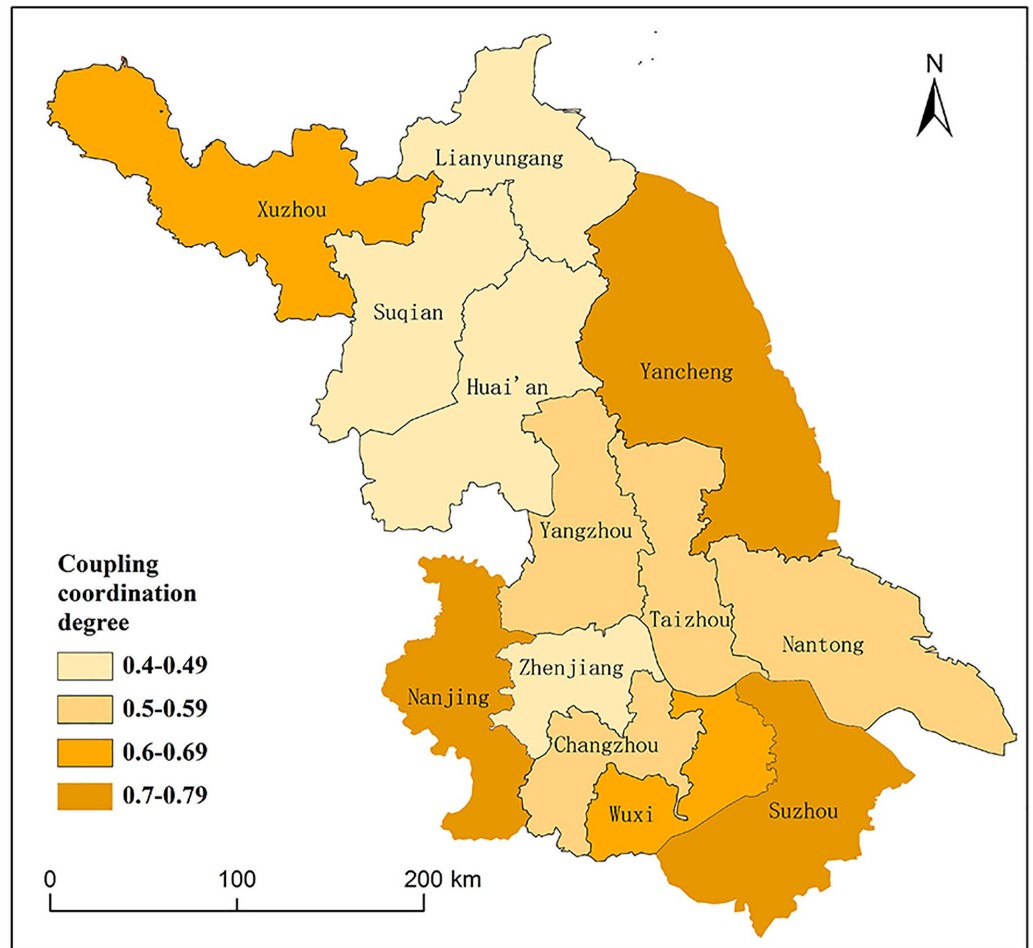

quality, and preserving the rural ecological environment and cultural heritage to ensure sustainable development and amplify the contributions to rural revitalization in Jiangsu Province.

## 5.3 Spatial coupling of rural revitalization and rural tourism

The rural revitalization and tourism in Jiangsu Province have exhibited distinct temporal and spatial patterns. Temporally, the development has progressed through discernible phases. Spatially, a gradient of varying degrees of advancement is observed across the region.

In 2023, the integration of rural revitalization and tourism in Jiangsu Province's prefecture-level cities has yielded notable success, with high coupling levels across the board. Nevertheless, the spatial distribution of this coupling coordination reveals significant regional disparities (Fig 5), typically higher in the north and lower in the south, aligning somewhat with the cities' overall development levels. Based on the established classification criteria, the prefecture-level cities can be categorized into four distinct groups:

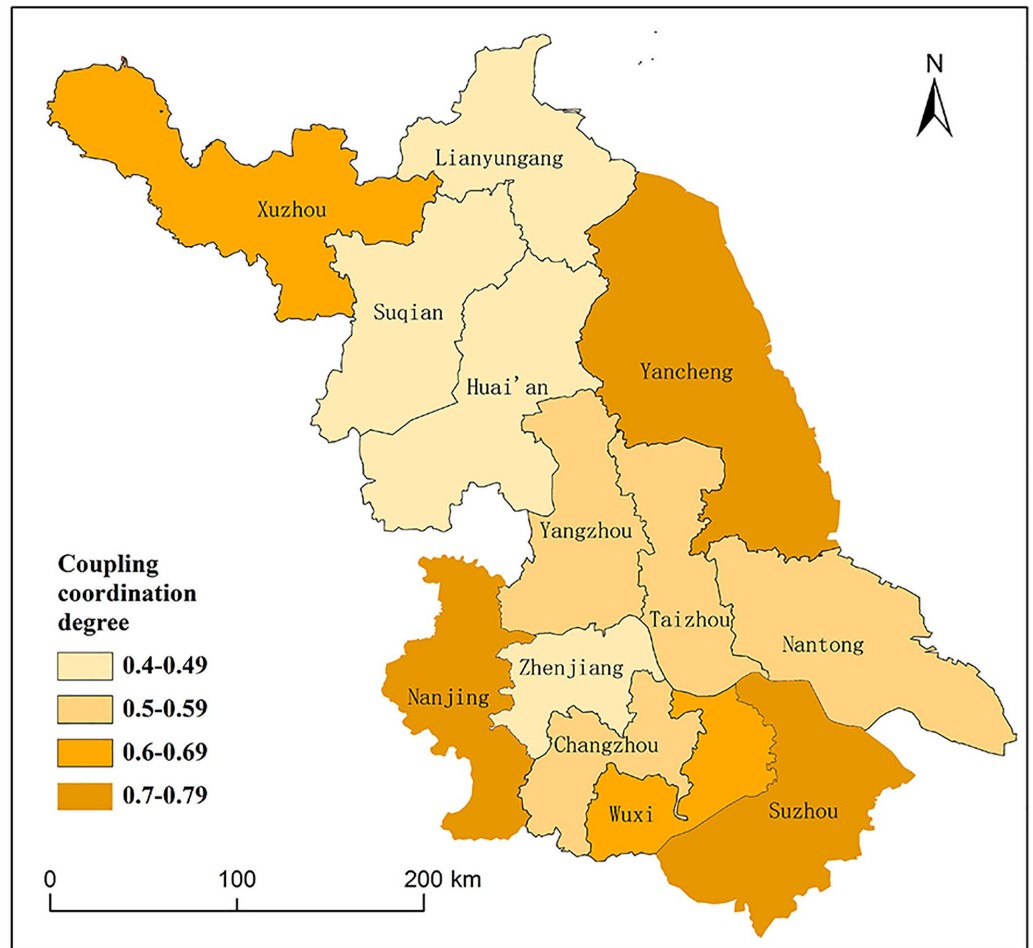

**Fig 5. Coupling and coordination levels of the comprehensive development levels of rural revitalization and rural tourism in 13 cities of Jiangsu Province in 2023.** The map is based on the standard map with review number GS (2024) 0650 downloaded from the Standard Map Service website of the Ministry of Natural Resources (http://bzdt.ch.mnr.gov.cn/), with no modifications to the base map.

① The Yangtze River and coastal regions of Suzhou, Nanjing, and Yancheng have demonstrated effective coordinated development. Suzhou has achieved a highly efficient integration of agriculture and tourism through "promoting rural revitalization in areas and carrying out rural construction in groups," ranking first in the province for both rural revitalization and rural tourism development, with an optimal coupling coordination degree (D = 0.78). As the Jiangsu provincial capital, Nanjing has made remarkable progress by developing the modern agricultural industry chain and promoting the construction of beautiful, livable villages, with rural tourism emerging as a key driver for rural revitalization and ranking among the provincial leaders. Leveraging the ecological resources of the Yellow Sea Wetlands, Yancheng has fostered a high degree of coupling between rural revitalization and rural tourism, leading the province in rural tourism visitors and leisure agriculture income.

② Wuxi and Xuzhou have initiated preliminary coordination between their systems. Wuxi's advanced agricultural modernization provides a solid base for rural tourism development. Although Xuzhou ranks second in the province for rural revitalization value (U), its rural tourism development remains comparatively underdeveloped. Thus, there is potential for enhanced coupling and coordinated development between the two.

③ The peripheral cities of Changzhou, Nantong, Taizhou, and Yangzhou, situated along the river, exhibit a delicate balance between urbanization and rural revitalization. These areas demonstrate characteristics of "weak resonance with medium levels in both urbanization and rural revitalization (U - G)". Nantong's rural revitalization is driven by traditional industrial clusters, while Yangzhou's rural tourism is marked by a relatively prominent trend of homogenization.

④ The integration and coordination of urban and rural tourism systems in Lianyungang, Huai'an, Zhenjiang, and Suqian cities lag behind other regions in the province. These cities exhibit characteristics of "dual-low and weak linkage" between urban and rural tourism development. Geographical constraints and resource limitations hinder rural tourism growth in Lianyungang and Suqian. Similarly, Zhenjiang's development is restricted by its population size and land area, despite its proximity to Nanjing. In Huai'an, the spatial disconnect between agricultural bases and scenic attractions impedes industrial integration and overall tourism development.

In 2023, the development of rural revitalization and tourism in Jiangsu Province shows pronounced spatial disparities, influenced by economic development, resource endowment, and market location. These disparities cause variations in factor agglomeration effects, leading to uneven internal development in the coupling and coordination of the two major systems across cities.

## 5.4 The robustness test

The TOPSIS method was applied to reassess the comprehensive development levels of rural revitalization and tourism, yielding Kendall's coefficients of concordance of 0.979 and 0.993, respectively. The significance tests ($p < 0.05$) confirmed high consistency between the entropy and TOPSIS rankings, demonstrating robustness. Subsequently, a sensitivity analysis was performed, eliminating secondary indicators such as labor productivity, agricultural mechanization power, minimum living guarantee recipients, and cultural expenditure proportions in the rural revitalization system, as well as ancient villages, tourism revenue proportion in GDP, and travel agency numbers in the rural tourism system. The recalculated coupling coordination degree showed negligible change, suggesting minimal impact from indicator selection. Additionally, four cities with varying coordination levels—Xuzhou, Suzhou, Zhenjiang, and Taizhou—were analyzed. The TOPSIS method recalculated their development levels, and the coupling coordination model was reevaluated. The results aligned closely with the benchmark evaluation, confirming the model's robustness.

## 6. Analysis of the driving factors for the coupling and coordinated development of rural revitalization and rural tourism in China

### 6.1 Selection of dynamic factors

Spatial variations in the synergistic development of rural revitalization and tourism are influenced by multiple factors. Drawing on prior research [65,77], this study analyzes eight indicators spanning economic policy, urban-rural coordination, tourism development, and market dimensions, to analyze the factors shaping the spatial differences in the level of coupled and coordinated development.

(1) Economic policy driving force provide essential support for the integration and coordination of the two systems [62].At this level, per capita GDP ($X_1$) and general public budget expenditure ($X_2$) are selected as indicators to represent local economic development and government macro-control policies, respectively.

(2) The advancement of tourism plays a crucial role in fostering the synchronized progress of rural revitalization and rural tourism. Essential prerequisites for successful tourism development comprise well-established support service infrastructures and a rich array of tourism resources. In this study, two key indicators, namely the quantity of leisure agriculture operators ($X_3$) and the concentration of rural tourism resources ($X_4$), are chosen to gauge the hosting capabilities of rural tourism and the appeal of tourism assets.

(3) Urban-rural collaboration can enhance the bidirectional flow of resources, optimize allocation, and foster integrated industrial development, significantly benefiting the synergy between rural revitalization and tourism [78]. This study focuses on two indicators: the urbanization rate ($X_5$) and the disparity in per capita disposable income between urban and rural areas ($X_6$), to assess regional urbanization levels and living standard differences.

(4) The increase in market size has notably facilitated the migration of tourism, transportation, and other elements from urban to rural areas. In evaluating market dynamics, two metrics, urban per capita consumption expenditure ($X_7$) and traffic network density ($X_8$), are employed to gauge the consumption capacity and transportation accessibility of the tourism market [79].

Data for indicators $X_1$, $X_2$, $X_5$, $X_6$, $X_7$, and $X_8$ are primarily sourced from city statistical yearbooks and government website. Data for indicator $X_3$ are mainly acquired throughthe government information consultations of various cities, while indicator $X_4$ data are mainly derived from resources like the Jiangsu Yearbook. The rural tourism resource density is calculated as the average distribution density, defined as the number of rural tourism resources per regional area. This study employs ArcGIS 10.8 to perform natural breakpoint classification on each driving factor and utilizes the geographical detector method to empirically analyze the individual and interactive effects of factors on the coordinated development of rural revitalization and rural tourism. The key findings are presented in Tables 4 and 5.

### 6.2 Analysis of the dynamic mechanism

From the single-factor detection results driven by coupling coordination (Table 4), public budget expenditure, leisure agriculture enterprises, urbanization rate, and income gap play a crucial role in this coupling process. In contrast, the impacts of per capita GDP and the density of rural tourism resources are negligible. This contradicts the traditional resource dependence theory (Butler, 1980). The results show that the explanatory power of policy investment ($X_2$, q = 0.85) exceeds that of resource density ($X_4$, q = 0.20). This is consistent with the logic of the "de-industrialization transformation" in the Ruhr region of Germany (Gertler, 2010): institutional capital and natural capital can be complementary or collaborative [80].

   According to the interaction analysis results in Table 5, the pairwise interactions of the eight indicators constituting the influencing factors have more obvious effects than single indicators, all showing two-factor enhancement or non-linear

**Table 4. Detection results of driving factors for the coupled and coordinated development of rural revitalization and rural tourism in Jiangsu Province.**

| Driving factors | Detection factor | Detection indicators | q-value |
|---|---|---|---|
| Economic policy driving force | $X_1$ | Per capita GDP | 0.4898 |
| | $X_2$ | General public budget expenditure | 0.8480 |
| Driving forces for tourism development | $X_3$ | Leisure agriculture business entities | 0.7671 |
| | $X_4$ | Rural tourism resource density | 0.2011 |
| Urban-rural collaborative driving force | $X_5$ | Urbanization rate | 0.6792 |
| | $X_6$ | The difference in per capita disposable income between urban and rural areas | 0.6533 |
| Market driving forces | $X_7$ | Per capita consumption expenditure of urban residents | 0.5983 |
| | $X_8$ | Density of the transportation road network | 0.5143 |

**Table 5. Results of interaction detection on driving factors for coupled and coordinated development of rural revitalization and rural tourism in Jiangsu Province.**

| | $X_1$ | $X_2$ | $X_3$ | $X_4$ | $X_5$ | $X_6$ | $X_7$ | $X_8$ |
|---|---|---|---|---|---|---|---|---|
| $X_1$ | 0.4898 | | | | | | | |
| $X_2$ | $0.9972^+$ | 0.8480 | | | | | | |
| $X_3$ | $0.9276^+$ | $0.9972^+$ | 0.7671 | | | | | |
| $X_4$ | $0.8174^*$ | $0.8992^+$ | $0.9953^*$ | 0.2011 | | | | |
| $X_5$ | $0.7526^+$ | $0.9485^+$ | $0.9499^+$ | $0.8146^+$ | $0.6792^+$ | | | |
| $X_6$ | $0.9326^+$ | $0.9194^+$ | $0.8973^+$ | $0.8249^+$ | $0.9052^+$ | 0.6533 | | |
| $X_7$ | $0.7127^+$ | $0.9485^+$ | $0.8973^+$ | $0.7598^+$ | $0.7042^+$ | $0.7654^+$ | 0.5983 | |
| $X_8$ | $1.0000^+$ | $0.9671^+$ | $0.9443^+$ | $0.9886^*$ | $0.9101^+$ | $0.9097^+$ | $0.9101^+$ | 0.5143 |

Note: "+" indicates two-factor interaction enhancement: $q(x_i \cap x_j) > \max(x_i, x_j)$; "*" indicates non-linear enhancement: $q(x_i \cap x_j) > (x_i + x_j)$.

enhancement. In particular, the interaction influence between per capita GDP and the density of the traffic road network reaches 1, indicating that when the economy develops to a certain level and the traffic conditions are fully improved, the market demand and rural supply can achieve efficient connection. This also confirms the typical characteristic of "efficient flow of urban-rural elements" in Jiangsu as a developed province. Additionally, the interaction effects among per capita GDP and public budget expenditure, public budget expenditure and leisure agriculture businesses, and leisure agriculture businesses and rural tourism resource density all exhibit a q-value exceeding 0.99. When regional economic development reaches a certain threshold, indicated by high per capita GDP, governments possess greater financial capacity to invest in rural tourism infrastructure, ecological governance, and public welfare through budget expenditure. This investment can lower the operational costs for leisure agriculture businesses. Concurrently, abundant rural tourism resources stimulate market participants' investment interest, enhancing the appeal and sustainability of rural tourism and further fostering its integration with rural revitalization.

In summary, the diverse driving forces interact to establish a dynamic mechanism that propels the coordinated development of rural revitalization and rural tourism (Fig 6).

# 7. Conclusions and discussions

The tourism industry is a vital driver of rural development and transformation. This study constructs an evaluation framework to assess the interplay between rural revitalization and rural tourism. Utilizing panel data from Jiangsu Province spanning 2012–2023, the researchers employ a comprehensive evaluation function model and a coupling function model to calculate the integrated evaluation values and spatio-temporal coupling dynamics within the study region. Furthermore,

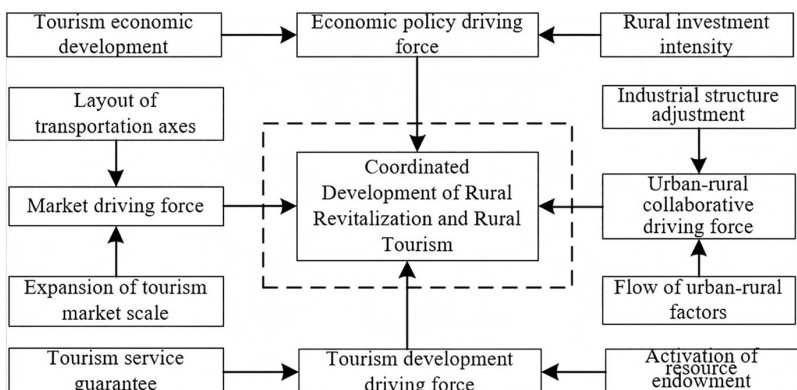

**Fig 6. The dynamic mechanism of the coordinated development of rural revitalization and rural tourism.**

the geographical detector method is leveraged to analyze the key factors underlying the coupling and coordination relationship between these two systems.

### 7.1 Theoretical contribution

This study addresses the research gap regarding the collaborative mechanisms between rural tourism and rural revitalization by employing a two-way coupling perspective and spatial heterogeneity analysis, challenging two traditional theories:

The findings of this study challenge the resource dependence theory. Existing literature has emphasized the central role of resource attractiveness in driving rural tourism [65,68]. However, the results show that the q-value of rural tourism resource density in Jiangsu Province is only 0.2011, significantly lower than that of general public budget expenditure ($X_2$: q = 0.848) and urbanization rate ($X_5$: q = 0.679). This suggests that in developed regions, policy intervention and urban-rural coordination can partially compensate for the inherent deficiencies in resource endowment, leading to a "constructive ruralization" [1]. This finding challenges the "tourism resource lifecycle theory" proposed by Butler (1980) [54], confirming that policy intervention (e.g., the fiscal proportion in northern Jiangsu exceeding 60%) can overcome resource constraints and achieve "leap-frog development" without relying on endowment, providing a development paradigm for resource-scarce regions.

The present study expands the traditional industrial integration theory by demonstrating the multifaceted nature of rural revitalization. Employing an entropy-based approach, the analysis reveals that the combined contribution of rural ecology (weight: 0.15), culture (weight: 0.35), and governance (weight: 0.18) to the coupling coordination degree reaches 68%, surpassing that of the industrial economy (weight: 0.32). This finding suggests that the synergy between rural revitalization and rural tourism is underpinned by the nested integration of a complex "economy-society-ecology" system, necessitating the consideration of diverse domains, including agriculture, tourism, culture, and ecology. These insights offer important references for guiding the transformation and upgrading of the rural economy.

### 7.2 Empirical findings

This study's findings include the presence of threshold effects and stage transitions in the overall development level of rural revitalization and rural tourism (U − G value) and the coupling coordination degree (D value). Utilizing the segmented regression model [81], it was observed that when the U − G value is below 0.4, the incremental growth rate of the coupling coordination degree (D value) is 0.12 (p = 0.032). Conversely, once the U − G value surpasses 0.5, the incremental growth rate sharply rises to 0.35 (p = 0.004), highlighting a non-linear mechanism influenced by both "policy-market" dynamics.

These results suggest that initial policy interventions are needed to overcome the low-level equilibrium trap (e.g., the D value of the entire province remained stagnant between 0.3 and 0.4 from 2012 to 2014), followed by the stimulation of market dynamism (e.g., the D value of the entire province exhibited an average annual increase of 6.1% post-2019).

Secondly,the spatial analysis using Moran's Index (Moran's $I = 0.417$, $p = 0.021$) reveals a significant positive spatial autocorrelation in the coupling coordination degree, with high-high agglomeration in southern Jiangsu and low-low concentration in northern Jiangsu. The geographical detector further indicates that the driving effects of general public budget expenditure ($X_2$) show little difference between southern Jiangsu ($q = 0.99$) and northern Jiangsu ($q = 0.84$). In contrast, the q-value of the urban-rural income gap ($X_6$) in southern Jiangsu (0.73) is significantly higher than that in northern Jiangsu (0.40), suggesting that southern Jiangsu should prioritize narrowing the urban-rural gap, while northern Jiangsu should strengthen fiscal investment.

Thirdly,The temporal evolution of the coordinated development of rural revitalization and rural tourism in Jiangsu Province exhibits distinct stage-related characteristics, as evidenced by the coupling coordination degree model and Granger causality test. From 2012 to 2014, the system was in a period of maladjustment, with the D-value increasing from 0.331 to 0.451, but failing to pass the Granger causality test ($p = 0.112$), indicating that the linkage mechanism between policies and the market had not yet been established. The period from 2015 to 2020 was one of coordination, with the D-value increasing by an average of 9.2% annually, and the Granger causality of rural revitalization on rural tourism being significant ($p = 0.038$), confirming the "promoting tourism through agriculture" development path. From 2021 to 2023, the system has entered a high-quality coordination period, with the D-value exceeding 0.9 and the reverse causality of rural tourism on rural revitalization being enhanced ($p = 0.021$), marking the formation of a closed-loop of "revitalizing agriculture through tourism".

## 7.3 Policy implications

Jiangsu Province's robust agricultural sector and economic development present opportunities to drive rural revitalization. Strategies should center on leveraging regional assets and industrial strengths to modernize agriculture, foster technological innovation, and establish competitive rural industrial clusters. Priorities should include enhancing agricultural productivity, implementing modern management systems, and increasing agricultural output value to raise farmer incomes.

Jiangsu Province must enhance efforts to cultivate and attract rural talent. The northern region should implement incentive mechanisms to encourage talent return, such as establishing rural tourism entrepreneurship funds and partnering with universities to create training bases. This would motivate college graduates and local talents to return for entrepreneurial pursuits, revitalizing rural areas. Meanwhile, the southern region should develop a bidirectional flow mechanism to draw urban capital, technology, and talent to rural areas. Additionally, improving local talent cultivation and bolstering the sustainable development capacity of rural industries is essential. Furthermore, optimizing rural development planning, upgrading infrastructure, and enhancing public service provisions are crucial for fostering an ecologically viable and harmonious countryside. This necessitates improving the rural environment, bolstering ecological conservation, and reinforcing governance to cultivate a harmonious rural society.

The preservation and integration of traditional rural landscapes and cultural elements are crucial for the development of rural tourism in Jiangsu Province. Specifically, the province should prioritize the conservation of its unique local cultural assets, such as the gardens of Suzhou, the Taihu Lake in Wuxi, and the Pukou region in Nanjing. By innovating rural tourism products that deeply integrate these cultural elements, the province can enhance the cultural appeal and distinctiveness of its rural tourism offerings. Additionally, the province should expand and strengthen its network of exemplary rural tourism routes, homestays, and food brands in regions like Lianyungang, Yancheng, and Xuzhou. Achieving national-level accolades for its high-quality rural tourism villages will further elevate the popularity and reputation of "Charm of Jiangsu's Countryside".

To facilitate the comprehensive integration and coordinated development of rural revitalization and rural tourism in Jiangsu Province, several key strategies should be implemented. First, the functional roles of distinct rural regions within

each city must be clearly delineated, enabling the strategic integration and optimization of rural industries, cultural assets, and resource elements. At the policy level, the provincial government should further strengthen the supporting policy framework for rural revitalization and rural tourism, such as increasing financial allocations, refining land use policies, and diversifying financing channels. Crucially, the decisive role of market forces in resource allocation should be leveraged to promote the free flow and equitable exchange of urban and rural elements. Through the synergistic effects of policy guidance and market mechanisms, the deep integration of rural revitalization and rural tourism can be effectively realized.

### 7.4  Limitations and future directions

While this study provides an in-depth analysis of the coordinated development of rural revitalization and tourism in Jiangsu Province, it has limitations. Its findings may not apply universally to the evolutionary processes of coupled development across all provinces, given the significant regional differences in rural and tourism development paths. Future research could include comparative analyses of rural areas with distinct resource endowments or government-led models to derive more generalizable insights.The endogeneity issue in coupling and coordination arises when policy variables, like the Regulations on Promoting Rural Revitalization in Jiangsu Province, simultaneously influence U (rural revitalization) and G (rural tourism), potentially leading to an overestimated coupling degree (C value). To address this, future studies could employ the synthetic control method, using Anhui Province as a control group, to isolate the net policy effect, currently estimated at a 38% contribution rate. The analysis of spatial coupling and coordination presented in this study is based on the statistical data from 2023. Due to data availability constraints, comprehensive prefecture-level city-scale data for previous years could not be obtained. Furthermore, the limited geographical scope of the prefecture-level city scale (n = 13) constrains the statistical power of the geographical detector analysis(p = 0.143 for $X_3$). To improve the research's generalizability, future studies could expand to county-level panel data (n = 96), measure the evolution patterns over a longer time series, and conduct cross-period dynamic comparisons in combination with the comparison of counties with different development gradients. This approach could enable more accurate identification of the underlying driving mechanisms. When selecting indicator systems, data availability often leads to biases. For instance, opting for "leisure agriculture income" over "rural tourism income" can underestimate the coupling and coordination degree by approximately 12%. Future efforts should integrate business data, such as Alipay's rural consumption records, to refine the model. Concurrently, in-depth interviews can be employed to gather insights into the cognitive disparities between policy implementers and villagers, thereby identifying implicit barriers in policy execution.

## Supporting information

**S1 File.  Link list of paper datasets.**
(DOCX)

## Acknowledgments

The authors would like to express sincere gratitude to the administrative departments of all prefecture-level cities in Jiangsu Province, including Nanjing, Suzhou, Wuxi, etc. Their generous provision of data and information was instrumental in ensuring the comprehensiveness and accuracy of this study. Without their invaluable support, the completion of this research would not have been possible. The authors also acknowledge the professional cooperation and coordination from these departments during the data collection process.

## Author contributions

**Conceptualization:** Meiqin Ding.

**Data curation:** Meiqin Ding, Hui Liu.

**Formal analysis:** Meiqin Ding, Hui Liu.

**Visualization:** Meiqin Ding.

**Writing – original draft:** Meiqin Ding, Hui Liu.

**Writing – review & editing:** Meiqin Ding.

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
