## [Decision Letter · Decision Letter 0]

23 Jul 2025

PONE-D-25-31895Research on the Spatiotemporal Coupling and Coordinated Development of Rural Tourism and Rural Revitalization: A Case Study of Jiangsu ProvincePLOS ONE

Dear Dr. Ding,

Thank you for submitting your manuscript to PLOS ONE. After careful consideration, we feel that it has merit but does not fully meet PLOS ONE’s publication criteria as it currently stands. Therefore, we invite you to submit a revised version of the manuscript that addresses the points raised during the review process.

Thank you for submitting your manuscript to PLOS ONE. After careful review, we are pleased to invite you to revise and resubmit your manuscript. The reviewers have provided helpful feedback, and with revisions, your work has the potential to meet the journal's standards for publication.

We look forward to receiving your revised submission. Thank you for considering PLOS ONE.

We look forward to receiving your revised manuscript.

Kind regards,

Chunguang Hu

Academic Editor

PLOS ONE

2. We note that Figures 3-4 in your submission contain [map/satellite] images which may be copyrighted. All PLOS content is published under the Creative Commons Attribution License (CC BY 4.0), which means that the manuscript, images, and Supporting Information files will be freely available online, and any third party is permitted to access, download, copy, distribute, and use these materials in any way, even commercially, with proper attribution. For these reasons, we cannot publish previously copyrighted maps or satellite images created using proprietary data, such as Google software (Google Maps, Street View, and Earth). For more information, see our copyright guidelines: http://journals.plos.org/plosone/s/licenses-and-copyright.

1. You may seek permission from the original copyright holder of Figures 3-4 to publish the content specifically under the CC BY 4.0 license. 

3. We notice that your supplementary figures are included in the manuscript file. Please remove them and upload them with the file type 'Supporting Information'. Please ensure that each Supporting Information file has a legend listed in the manuscript after the references list.

Additional Editor Comments (if provided):

Reviewers' comments:

Reviewer's Responses to Questions

**Comments to the Author**

1. Is the manuscript technically sound, and do the data support the conclusions?

Reviewer #1: Yes

Reviewer #2: Partly

Reviewer #3: Partly

2. Has the statistical analysis been performed appropriately and rigorously? 

Reviewer #1: Yes

Reviewer #2: Yes

Reviewer #3: No

3. Have the authors made all data underlying the findings in their manuscript fully available?

Reviewer #1: Yes

Reviewer #2: No

Reviewer #3: No

4. Is the manuscript presented in an intelligible fashion and written in standard English?

Reviewer #1: Yes

Reviewer #2: Yes

Reviewer #3: No

5. Review Comments to the Author

Reviewer #1: This manuscript presents a comprehensive study on the spatiotemporal coupling and coordinated development of rural tourism and rural revitalization in Jiangsu Province, using panel data from 2012 to 2023. The methods are rigorous, and the results are supported by solid data. The presentation is clear. The manuscript is recommended for publication in PLOS ONE. Some minor revisions are suggested below.

1. Is there any correlation between driving factors?

2. Will different models produce the same results?

3. How is “rural tourism resource density” defined?

4. The authors give differentiated policy recommendations: northern Jiangsu should focus

on talent attraction and the integration of culture and tourism, while southern Jiangsu should explore mechanisms to facilitate the two-way flow of urban-rural elements. The problem is how could northern Jiangsu attract more talents?

5. Besides data, some surveys, which may reflect some hidden reasons, are highly suggested.

6. There are some typos in the manuscript.

Reviewer #2: Overall comments:

The manuscript explores the spatiotemporal coupling and coordinated development of rural tourism and rural revitalization in Jiangsu Province, utilizing a combination of entropy-based weighting, coupling coordination degree models, and geographical detectors. This is a timely and policy-relevant topic that adds value to the academic discourse surrounding rural transformation and tourism development, particularly in the context of developed regions in China. The topic is relevant and potentially significant in understanding regional development dynamics, especially in the context of developed regions undergoing rural transformation. The manuscript demonstrates a strong empirical foundation and introduces a critical reflection on classical theories such as resource dependence and industrial integration.

However, several aspects of the manuscript require substantial revision before it can be considered for publication.

1. Title and Abstract

• What Needs Improvement:

The title is overly long and could be more concise without losing clarity. The abstract is informative but densely packed with technical detail (e.g., p-values, q-values) that may not be suitable for a general audience.

• Recommendation:

Simplify the title while retaining key terms. Revise the abstract to follow a clearer structure (objective, methods, key results, implications), avoid excessive statistics, and highlight the study’s unique contribution.

2. Introduction

• Strength:

The introduction provides a relevant overview of urbanization and the need for rural revitalization in China.

• What Needs Improvement:

Some of the opening statements are too generic. The research gap is not clearly articulated until later in the section.

• Recommendation:

Clarify the problem statement and theoretical gaps earlier in the introduction. Frame the study’s objective more precisely to distinguish it from past research.

3. Literature Review

• Strength:

The review covers both domestic and international studies and discusses relevant theoretical perspectives.

• What Needs Improvement:

The section is overly descriptive and reads more like an annotated bibliography than a synthesis.

• Recommendation:

Reorganize this section thematically. Focus more on identifying what is not known rather than summarizing what is already established. Include a conceptual framework to clearly justify the study design.

4. Methodology

• Strength:

The methods employed (entropy method, coupling coordination degree model, and geographical detector) are appropriate and robust.

• What Needs Improvement:

The methodological section is lengthy, technical, and lacks interpretive clarity. There is no sensitivity analysis or explanation of why specific indicators were chosen over others.

• Recommendation:

Condense and clarify the methodological descriptions. Justify the choice of indicators and models with supporting literature. Consider adding robustness checks or model validation.

5. Results and Discussion

• Strength:

The results are comprehensive and presented with supporting figures and tables.

• What Needs Improvement:

The discussion is largely descriptive and does not critically engage with previous literature or explore the implications of the findings in depth.

• Recommendation:

Enhance the analytical depth of the discussion by comparing results with past studies. Avoid overuse of numerical data in text and focus more on interpretation and significance.

6. Conclusions and Policy Implications

• Strength:

The conclusions present relevant recommendations for regional development and tourism policy.

• What Needs Improvement:

The theoretical claims (e.g., revision of resource dependence theory) are ambitious and need stronger empirical validation.

• Recommendation:

Temper broad theoretical assertions unless backed by comparative evidence. Emphasize practical contributions and highlight potential applications beyond Jiangsu Province.

7. Data and Reproducibility

• What Needs Improvement:

Not all data are openly available; some are obtained through government correspondence, limiting reproducibility.

• Recommendation:

Clearly specify which data are publicly available and which are not. If possible, provide anonymized datasets or aggregated indicators for reproducibility.

8. Limitations and Future Research

• Strength:

The authors acknowledge several important limitations.

• What Needs Improvement:

Some limitations, such as the selection bias in indicators and restricted data scope, could be further elaborated.

• Recommendation:

Strengthen this section by discussing how future comparative or longitudinal studies (e.g., using county-level data or other provinces) could improve the generalizability of findings.

Reviewer #3: This study uses the entropy method, coupling coordination degree model, and geographical detector to elucidate the nonlinear dynamics and driving factors of their spatio-temporal evolution in Jiangsu province in China from 2012 to 2023.

1. The abstract should be reorganized. The contents of main findings should be reduced, and those of methodologies and research contents in this manuscript should be supplemented.

2. In Line 27 ‘.037 and .156.’, it should add zero before decimal point.

3. The international background of the study should be provided in Section Introduction.

4. The importance of the study should be mentioned in Section Introduction.

5. The research gap should be identified to show the novelty of this study. Section 2.4 cannot show the research gap in current research progress, the novelty of this study and contribution to the theories.

6. Why the indicators are selected in Section 3.2 and Table 1? There is no explanation. The evidence or citations should be provided.

7. Section 3.3 is too simple. The data collection progress, data cleaning and preprocess are missing. Which indicator is from Jiangsu Statistical Yearbook, and which is from Jiangsu Tourism Yearbook, statistical yearbooks of local cities, or relevant government websites? Where individual indicator data were unavailable, historical statistical materials and reports were used to infer the values. How to infer the unavailable data? This section cannot show the reliability of the data, which may affect the results.

8. The standard for the classification of coupling degree C and coupling coordination degree D should be cited.

9. Why the indicators are selected in Table 3? The reasons and explanations should be provided. The data source of these variables should be given.

10. The analysis of Section 6.2 should be discussed in more depth combined with policies and development level of the variables.

11. The clarity of all figures should be improved.

12. There are some Chinese parentheses in this manuscript. Please have a check.

6. PLOS authors have the option to publish the peer review history of their article (what does this mean? ). If published, this will include your full peer review and any attached files.

**Do you want your identity to be public for this peer review?** For information about this choice, including consent withdrawal, please see our Privacy Policy .

Reviewer #1: No

Reviewer #2: **Yes: ** MICHAEL CHRISTIAN

Reviewer #3: No

---

## [Author Response · Author response to Decision Letter 1]

9 Sep 2025

To: PLOS ONE Editorial Department

Subject: Revised manuscript resubmission(Response to manuscript PONE-D-25-31895)

Dear Editor ChunguangHu and PLOS ONE Reviewer:

We appreciate the opportunity to revise our manuscript titled "Spatiotemporal Coupling and Coordinated Development of Rural Revitalization and Rural Tourism in Jiangsu" and are grateful for the insightful comments provided by the reviewers. Those comments are all valuable and very helpful for revising and improving our paper, as well as the important guiding significance to our researches.

We have thoroughly revised the manuscript in accordance with your suggestions. Enclosed with this correspondence are:

1. A manuscript marked with all the changes ("Revised Manuscript with Track Changes") ;

2. A clean version of the revised manuscript("Manuscript");

3. This reply letter("Response to Reviewers").

In the following, we have provided detailed responses to each of the reviewers' comments. Revised portion are marked in red in the paper. Additionally, we have conducted a comprehensive revision of the entire manuscript. In this response letter, the reviewers' comments are presented in italics, and our corresponding changes and additions to the manuscript are highlighted in red text. We have tried our best to make all the revisions clear, and we hope that the revised manuscript meets the requirements for publication.

Part I: Responding to the Journal's Requirements

We have ensured that the manuscript fully complies with the formatting requirements of PLOS ONE and have processed all the files as required.

1. Regarding the copyright of Figure (Figure Number)

Response 1: The map used in the paper is based on the standard map with review number GS (2024) 0650 downloaded from the Standard Map Service website of the Ministry of Natural Resources (http://bzdt.ch.mnr.gov.cn/), with no modifications to the base map. For the proof of the granted permission, please refer to the attached file"Proof of granted permissions".

2. Regarding the supporting information:

Response 2� We have moved the supplementary figures in the original manuscript to a separate "Supporting Information" file and ensured that each supporting information file has a a legend listed in the manuscript after the references list.

Part II: Point-by-point Response to the Reviewers' Comments

Reviewer #1:This manuscript presents a comprehensive study on the spatiotemporal coupling and coordinated development of rural tourism and rural revitalization in Jiangsu Province, using panel data from 2012 to 2023. The methods are rigorous, and the results are supported by solid data. The presentation is clear. The manuscript is recommended for publication in PLOS ONE. Some minor revisions are suggested below.

Response : Thank you very much for your strong support of our work, you have provided us with very valuable advice to improve the quality of this paper! We have used your commen to revise the manuscript and have attached a point-by-point response to your comments:

Comment 1: Is there any correlation between driving factors?

Response 1: This is a very good question. We did take into account the possible interactive relationships among the driving factors. We additionally employed the interaction detection function of the geographical detector, focusing on analyzing the effects of the combined action of multiple factors. The findings are presented in Table 5, accompanied by a detailed analysis of their correlations.

According to the interaction analysis results in Table 5, the pairwise interactions of the eight indicators constituting the influencing factors have more obvious effects than single indicators, all showing two-factor enhancement or non-linear enhancement. In particular, the interaction influence between per capita GP and the density of the traffic road network reaches 1, indicating that when the economy develops to a certain level and the traffic conditions are fully improved, the market demand and rural supply can achieve efficient connection. This also confirms the typical characteristic of "efficient flow of urban-rural elements" in Jiangsu as a developed province. Additionally, the interaction effects among per capita GDP and public budget expenditure, public budget expenditure and leisure agriculture businesses, and leisure agriculture businesses and rural tourism resource density all exhibit a q-value exceeding 0.99. When regional economic development reaches a certain threshold, indicated by high per capita GDP, governments possess greater financial capacity to invest in rural tourism infrastructure, ecological governance, and public welfare through budget expenditure. This investment can lower the operational costs for leisure agriculture businesses. Concurrently, abundant rural tourism resources stimulate market participants' investment interest, enhancing the appeal and sustainability of rural tourism and further fostering its integration with rural revitalization.

(See the second paragraph of section "6.2 Analysis of the dynamic mechanism")

Comment 2: Will different models produce the same results?

Response 2: Thank you for highlighting this. To verify the robustness of the results of our main model (coupling coordination degree model), we followed your suggestion and conducted additional sensitivity analyses. The results have been added to Section 5.4.

5.4 The robustness test

The TOPSIS method was applied to reassess the comprehensive development levels of rural revitalization and tourism, yielding Kendall's coefficients of concordance of 0.979 and 0.993, respectively. The significance tests (p < 0.05) confirmed high consistency between the entropy and TOPSIS rankings, demonstrating robustness. Subsequently, a sensitivity analysis was performed, eliminating secondary indicators such as labor productivity, agricultural mechanization power, minimum living guarantee recipients, and cultural expenditure proportions in the rural revitalization system, as well as ancient villages, tourism revenue proportion in GDP, and travel agency numbers in the rural tourism system. The recalculated coupling coordination degree showed negligible change, suggesting minimal impact from indicator selection. Additionally, four cities with varying coordination levels—Xuzhou, Suzhou, Zhenjiang, and Taizhou—were analyzed. The TOPSIS method recalculated their development levels, and the coupling coordination model was reevaluated. The results aligned closely with the benchmark evaluation, confirming the model's robustness.

(See the section "5.4 The robustness test")

Comment 3: How is “rural tourism resource density" defined?

Response 3: Thank you for pointing this out. We should define this core indicator more clearly in the paper. We have provided the definitions.

Data for indicators X1, X2, X5, X6, X7, and X8 are primarily sourced from city statistical yearbooks and government website. Data for indicator X3 are mainly acquired throughthe government information consultations of various cities, while indicator X4 data are mainly derived from resources like the Jiangsu Yearbook. The rural tourism resource density is calculated as the average distribution density, defined as the number of rural tourism resources per regional area.

(See the last paragraph of section "6.1 Selection of dynamic factors")

Comment 4: The authors give differentiated policy recommendations: northern Jiangsu should focus on talent attraction and the integration of culture and tourism, while southern Jiangsu should explore mechanisms to facilitate the two-way flow of urban-rural elements. The problem is how could northern Jiangsu attract more talents?

Response 4: Thank you for raising this profound policy issue. We have conducted more in - depth thinking and improvement on it. The specific solutions are detailed in the"7.3 Policy Implications"section .

Jiangsu Province must enhance efforts to cultivate and attract rural talent. The northern region should implement incentive mechanisms to encourage talent return, such as establishing rural tourism entrepreneurship funds and partnering with universities to create training bases. This would motivate college graduates and local talents to return for entrepreneurial pursuits, revitalizing rural areas.

(See the second paragraph of section "7.3 Policy Implications")

Comment 5: Besides data, some surveys, which may reflect some hidden reasons, are highly suggested.

Response 5: We fully agree with your view that qualitative research can better reveal the underlying mechanisms. Since this study is mainly a quantitative analysis based on macro panel data, it is not feasible to incorporate new field surveys in this revision. However,we have proposed it as a future research direction.Your suggestion has greatly helped improve the study's rigor.

Future efforts should integrate business data, such as Alipay's rural consumption records, to refine the model. Concurrently, in-depth interviews can be employed to gather insights into the cognitive disparities between policy implementers and villagers, thereby identifying implicit barriers in policy execution.

(See the section"7.4 Limitations and Future Directions"

Comment 6: There are some typos in the manuscript.

Response 6: Thank you for your careful review. We apologize for our carelessness. We have conducted multiple rounds of careful proofreading on the entire manuscript and corrected all the spelling and grammatical errors that were found.

Reviewer #2:The manuscript explores the spatiotemporal coupling and coordinated development of rural tourism and rural revitalization in Jiangsu Province, utilizing a combination of entropy-based weighting, coupling coordination degree models, and geographical detectors. This is a timely and policy-relevant topic that adds value to the academic discourse surrounding rural transformation and tourism development, particularly in the context of developed regions in China. The topic is relevant and potentially significant in understanding regional development dynamics, especially in the context of developed regions undergoing rural transformation. The manuscript demonstrates a strong empirical foundation and introduces a critical reflection on classical theories such as resource dependence and industrial integration.

Response : We sincerely appreciate your detailed, pertinent, and highly constructive comments, which have significantly enhanced the depth and breadth of this study. We have used your commen to revise the manuscript and have attached a point-by-point response to your comments:

Comment 1:Title and Abstract

• What Needs Improvement:

The title is overly long and could be more concise without losing clarity. The abstract is informative but densely packed with technical detail (e.g., p-values, q-values) that may not be suitable for a general audience.

• Recommendation:

Simplify the title while retaining key terms. Revise the abstract to follow a clearer structure (objective, methods, key results, implications), avoid excessive statistics, and highlight the study’s unique contribution.

Response 1: Thank you for your suggestion. We have streamlined the title to emphasize the core content and revised the abstract to enhance its structural clarity by delineating the purpose, methodology, key findings, and conclusion. We have minimized technical intricacies and emphasized the fundamental research contributions.

The revised title is as follows:

Spatiotemporal Coupling and Coordinated Development of Rural Revitalization and Rural Tourism in Jiangsu

The revised abstract now reads:

Rural tourism is pivotal in addressing the unidirectional urban-to-rural flow of resources, such as labor migration. However, the interaction between rural tourism and rural revitalization in developed regions remains poorly understood. This study establishes an evaluation index system for rural revitalization and rural tourism, examining their interrelationship. Using the entropy method and coupling coordination degree model, we assess the development levels and coordination degrees of these aspects in Jiangsu Province from 2012 to 2023. Furthermore, the geographical detector model is utilized to pinpoint the primary drivers influencing this coordination. The findings are: (1) Both rural revitalization and tourism exhibit significant growth, with southern Jiangsu outperforming the north; (2) The coupling coordination between these systems has strengthened, indicating a profound symbiotic relationship; (3) Spatial distribution differences are notable, with the coupling coordination degree D value in southern Jiangsu being 26.4% higher than in the north. This disparity is primarily attributable to the wider urban-rural income gap and greater fiscal investment in southern Jiangsu. Notably, the traditional "resource dependence theory" appears ineffective in Jiangsu, as the density of rural tourism resources is relatively low. Accordingly, the study proposes differentiated policy recommendations: northern Jiangsu should focus on talent attraction and the integration of culture and tourism, while southern Jiangsu should explore mechanisms to facilitate the two-way flow of urban-rural elements. This research provides a theoretical framework for coordinating "policy-market" dynamics in the rural transformation of developed regions.

(see Abstract

Comment 2: Introduction

• Strength:

The introduction provides a relevant overview of urbanization and the need for rural revitalization in China.

• What Needs Improvement:

Some of the opening statements are too generic. The research gap is not clearly articulated until later in the section.

• Recommendation:

Clarify the problem statement and theoretical gaps earlier in the introduction. Frame the study’s objective more precisely to distinguish it from past research.

Response2 :Thank you for your feedback. We have significantly revised the introduction section by clearly articulating the research question and theoretical gap at the outset. Additionally, we have provided a more precise delineation of the study's objectives and highlighted its distinctions from prior research.

The refined content is as follows:

Amid the growing focus on global urban-rural integration and rural revitalization strategies, rural tourism has emerged as a pivotal component in advancing rural revitalization[1], has attracted great attention from policymakers. Currently, the academic community has widely recognized the interrelationship between rural revitalization and rural tourism, examining the latter's development within the context of the former . However, existing studies mostly rely on interface data, making it difficult to capture dynamic evolution. Notably, the quantitative analysis of high-density ruralization's impact on the coupling mechanism in developed regions remains unexplored. Thus, a thorough investigation into achieving the seamless integration of these elements is urgently required.

With the acceleration of the global urbanization process, a large - scale migration of the rural population to cities has occurred. Villages see labour shortages, recession and social degradation. In most regions, countryside communities are becoming less viable[2].

(See the first paragraph of section "1 Introduction")

Comment 3: Literature Review

• Strength:

The review covers both domestic and international studies and discusses relevant theoretical perspectives.

• What Needs Improvement:

The section is overly descriptive and reads more like an annotated bibliography than a synthesis.

• Recommendation:

Reorganize this section thematically. Focus more on identifying what is not known rather than summarizing what is already established. Include a conceptual framework to clearly justify the study design.

Response3 :Thank you for your expert guidance. The "2. Literature Review" section has been revised to follow a thematic structure, organized as "established consensus→identified research gap→positioning of the current study". A concise overview of the research gap and the positioning of this paper is provided in Section 2.4. Additionally, a schematic diagram has been included to visually represent the theoretical framework and methodology of this study, thereby augmentin

---

## [Decision Letter · Decision Letter 1]

24 Sep 2025

Spatiotemporal Coupling and Coordinated Development of Rural Revitalization and Rural Tourism in Jiangsu

PONE-D-25-31895R1

Dear Dr. Ding,

We’re pleased to inform you that your manuscript has been judged scientifically suitable for publication and will be formally accepted for publication once it meets all outstanding technical requirements.

Kind regards,

Chunguang Hu

Academic Editor

PLOS ONE

Additional Editor Comments (optional):

Reviewer #2:

Reviewers' comments:

Reviewer's Responses to Questions

**Comments to the Author**

1. If the authors have adequately addressed your comments raised in a previous round of review and you feel that this manuscript is now acceptable for publication, you may indicate that here to bypass the “Comments to the Author” section, enter your conflict of interest statement in the “Confidential to Editor” section, and submit your "Accept" recommendation.

Reviewer #2: All comments have been addressed

2. Is the manuscript technically sound, and do the data support the conclusions?

Reviewer #2: Yes

3. Has the statistical analysis been performed appropriately and rigorously? 

Reviewer #2: Yes

4. Have the authors made all data underlying the findings in their manuscript fully available?

Reviewer #2: Yes

5. Is the manuscript presented in an intelligible fashion and written in standard English?

Reviewer #2: Yes

6. Review Comments to the Author

Reviewer #2: The authors have undertaken substantial revisions that significantly strengthen the manuscript. The abstract is now clearer and less technical, the introduction frames the research gap earlier and in a global context, the literature review is thematically reorganized with a conceptual framework, and methodological rigor has been enhanced through indicator justification and robustness checks. The discussion engages more critically with prior studies, theoretical claims have been moderated, and limitations are more thoroughly acknowledged with constructive directions for future research. Minor improvements are still possible in condensing descriptive literature review passages, simplifying figures, and more explicitly emphasizing the novelty of theoretical contributions, but these do not detract from the overall quality.

7. PLOS authors have the option to publish the peer review history of their article (what does this mean? ). If published, this will include your full peer review and any attached files.

**Do you want your identity to be public for this peer review?** For information about this choice, including consent withdrawal, please see our Privacy Policy .

Reviewer #2: **Yes: ** MICHAEL CHRISTIAN

---

## [Editor Report · Acceptance letter]

PONE-D-25-31895R1

PLOS ONE

Dear Dr. Ding,

I'm pleased to inform you that your manuscript has been deemed suitable for publication in PLOS ONE. Congratulations! Your manuscript is now being handed over to our production team.

Kind regards,

on behalf of

Dr. Chunguang Hu

Academic Editor

PLOS ONE